

SciPost Phys. Lect.Notes 41 (2022)

# A brief introduction to extended gravity and connections to dark energy: Illustrated with scalar field examples

**Clare Burrage**

School of Physics and Astronomy, University of Nottingham,
University Park, Nottingham NG7 2RD, United Kingdom

Clare.Burrage@nottingham.ac.uk

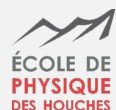

*Part of the Dark Matter*
*Session 118 of the Les Houches School, July 2021*
*published in the Les Houches Lecture Notes Series*

## Abstract

These lecture notes provide a brief introduction to extensions of gravity and their connections to dark energy. Due to time and space limitations this is not a comprehensive review of the field. Instead, I aim to introduce key concepts and ideas in this area through a series of examples based on scalar field extensions of gravity and models of dark energy.

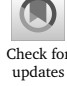

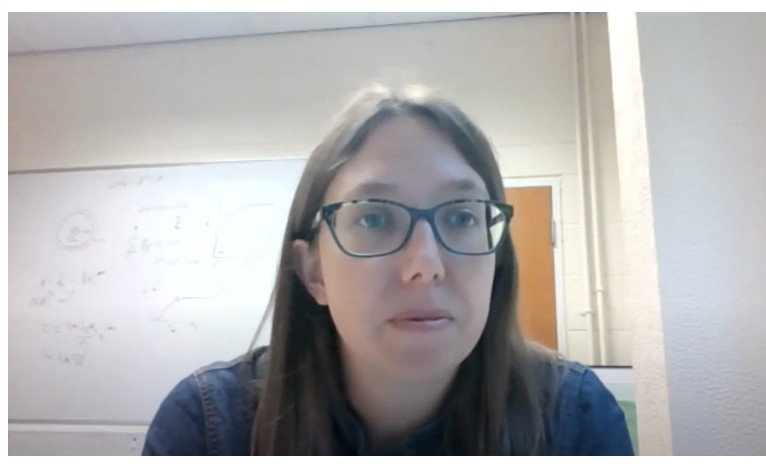

## 1   Introduction

The cosmological standard model contains a cosmological constant, $\Lambda$. The energy density associated with this component makes up almost 70% of the energy density in the universe today. In these lectures we will see how difficult it is to naturally encorporate this into our standard models of gravity and quantum mechanics. Dark energy is the name we give to the substance, fields or modifications of standard physics that solves this puzzle (although some authors prefer to use the name dark energy specifically for a field which reproduces all of the observed phenomenology).

In 2011 the Nobel prize was awarded to Saul Perlmutter, Brian Schmidt and Adam Riess for their observations of Type 1a supernovae. These exploding stars have approximately the same luminosity wherever and whenever they occur in the universe, and therefore can be used as standard(-izeable)[1] candles to form a cosmological distance ladder and help us reconstruct the recent expansion history of the universe. These observations [2, 3] (combined with observations of the CMB [4]) conclusively proved that the universe could not only be filled with

---

[1] The luminosity of Type 1a supernovae is not perfectly uniform, but a normalised version can be infered taking other observed properties of the supernova into account [1].

matter and radiation. It needs, in addition, a significant component of something that looks like a cosmological constant.[2] Before these observations there were already a number of indicators for a significant cosmological constant, including a tension between the ages of the oldest stars and the calculated age of the universe (in the absence of a cosmological constant), the number densities of galaxies, and the observed flatness of the universe [6].

The cosmological constant problem, is the difficulty of reconciling the observed value of the cosmological constant with our understanding of the Standard Model and its description in quantum field theory. As we will discuss in Section 4 there are both classical and quantum aspects of this problem. There are no widely accepted solutions to the cosmological constant problem; in this way the problem of dark energy is different to that of dark matter, where a number of different possible candidates have been identified, and the challenge is in identifying which option the universe prefers. Solutions to the cosmological constant problem [6–9] have been suggested within string theory based on anthropic principles, through modifications of gravity and and also by modifying our fundamental physical principles such as locality. It has also been suggested that we should just accept that the cosmological constant is fine tuned [10, 11].

In these lectures I will focus in particular on the possibility that the explanation for how the universe is evolving is due to a modification of gravity. Gravity has been extremely well tested in the laboratory, and in the solar system [12]. But it is an extrapolation over a vast range of scales to assume the same theory applies on cosmological scales. Therefore, it is possible that the solution to the cosmological constant problem is that the theory of gravity we use in our calculations is not correct on cosmological scales. It is important to note though, that the distinction between what is a modification of gravity and what is the introduction of novel matter is sometimes rather arbitrary.[3] Rather than debating classifications, we should focus on what degrees of freedom are present, what couples to what at what scale, and what the observational consequences are.

In these lectures I aim to demonstrate the problems associated with the cosmological constant and some of the phenomenology associated with proposed approaches to solving these problems. Rather than trying to be comprehensive I will use illustrative scalar field models through-out. New physics often means new particles - and scalars are the simplest option (especially if we don't have a reason to introduce direction or spin dependence). Examples that introduce new scalars include $f(R)$ modified gravity, massive gravity and quintessence models of dark energy. Because we are looking for new physics on longer distance scales in the universe these scalar fields are typically light.

I use the $(-, +, +, +)$ metric convention, except in Section 8.1 where we use the particle physicist's $(+, -, -, -)$ convention.

Key references used in putting together these notes are: [6], [7], [8] and [9].

## 2 General Relativity is Special

General relativity is our current best theory of gravity, it can be equivalently thought of as the theory of a curved space-time manifold, or as the theory of a massless spin two field. Lovelock's theorem is one way of expressing the uniqueness of General Relativity, as it limits the theories that one can construct from the metric tensor alone. For a full discussion of the uniqueness of general relativity we refer the reader to Ref. [7]. Lovelock's Theorem states that in a four

---

[2]Note that there is debate about whether these observations conclusively show the acceleration of the expansion, see for example Ref. [5].

[3]There are modifications of gravity that cannot be mapped to novel matter and vice versa, but there are also many theories which could be put into either category depending on how the theory is formulated [13].

dimensional space-time the only second-order equations of motion obtained from an action of the form

$$S = \int d^4x \mathcal{L}(g_{\mu\nu}) \tag{1}$$

are

$$\alpha\sqrt{-g}\left(R^{\mu\nu} - \frac{1}{2}g^{\mu\nu}R\right) + \lambda\sqrt{-g}g^{\mu\nu} = 0\,, \tag{2}$$

where $\alpha$ and $\lambda$ are constants. The simplest choice of action that gives such an equation of motion is

$$S = \int d^4x\sqrt{-g}\left(\alpha\frac{R}{2} - \lambda\right). \tag{3}$$

Second order equations of motion are required to ensure that no ghost-modes are present [14, 15].

If we include additional matter fields into this action, it must be done in a coordinate independent way, which means it must have the form

$$S = \int d^4x\sqrt{-g}\left(\alpha\frac{R}{2} - \lambda + \mathcal{L}_m(g_{\mu\nu}, \psi_i)\right). \tag{4}$$

Variation of equation (4), and appropriate choice of the constants $\alpha$ and $\lambda$, gives the Einstein equations

$$R_{\mu\nu} - \frac{1}{2}Rg_{\mu\nu} + \Lambda g_{\mu\nu} = M_p^{-2}T_{\mu\nu}\,, \tag{5}$$

where the stress-energy tensor of matter is

$$T_{\mu\nu} = -\frac{2}{\sqrt{-g}}\frac{\partial(\sqrt{-g}\mathcal{L}_m)}{\partial g^{\mu\nu}}\,. \tag{6}$$

The reduced Planck mass is defined to be $M_P^2 = 8\pi G$, where $G$ is Newton's constant. Note that the contracted Bianchi identities give the continuity equation

$$\nabla_\mu T^{\mu\nu} = 0\,. \tag{7}$$

## 3 Friedmann and Conservation Equations

Imposing the cosmological principles of homogeneity and isotropy, the Einstein equations become the Friedmann equations. If the scale factor of the universe is $a$, so that the Hubble 'constant' is $H = \dot{a}/a$. The Friedmann equations are then

$$H^2 = \frac{8\pi G}{3}\rho - \frac{K}{a^2} + \frac{\Lambda}{3}\,, \tag{8}$$

$$\frac{\ddot{a}}{a} = -\frac{4\pi G}{3}(\rho + 3p) + \frac{\Lambda}{3}\,, \tag{9}$$

where $\rho$ and $p$ are respectively the energy density and pressure of matter fields, which we have assumed can be modeled as a perfect fluid. From now on we are going to set $K$, the curvature of the universe, to be zero. The conservation equation for matter[4] is

$$\dot{\rho} + 3H(\rho + p) = 0\,. \tag{10}$$

---

[4]Note that this is not independent of the Friedmann equations, and can be obtained by differentiating and combining equations (8) and (9).

If the universe is dominated by a cosmological constant then

$$H = \sqrt{\frac{\Lambda}{3}}, \tag{11}$$

and

$$a \propto e^{\sqrt{\Lambda/3}t}, \tag{12}$$

so we see that the cosmological constant drives an accelerated expansion. We can also see from equation (9) that when $\Lambda$ is large enough the second derivative of the scale factor is positive, so the expansion accelerates.

Cosmological observations [4] indicate that our universe is currently dominated by a cosmological constant with a value

$$\Lambda = (2 \times 10^{-33} \text{ eV})^2. \tag{13}$$

We can also ask what properties a substance would have to have to mimic this expansion. From equation (9) we see that this means imposing the unusual requirements

$$\rho = \frac{\Lambda}{8\pi G}, \tag{14}$$

and $p = -\rho$ (equivalently the equation of state of the matter fluid should be $w = -1$). For matter to mimic the observed cosmological constant it would need to have an energy density

$$\rho \sim (2 \times 10^{-3} \text{ eV})^4. \tag{15}$$

## 4 The Problems of the Cosmological Constant

The stress-energy of matter in vacuum has to be of the form

$$\langle 0|T_{\mu\nu}|0\rangle = -\rho_{\text{vac}}g_{\mu\nu}, \tag{16}$$

where the terms on the right hand side are $g_{\mu\nu}$ so that the vacuum is Lorentz invariant (observations indicate that the vacuum should be invariant under arbitrary spatial rotations and should be identical as seen by observers moving relative to each other at constant speed) and $\rho_{\text{vac}}$ is a constant so that the stress energy is conserved.

The observed cosmological constant is the sum of a bare cosmological constant in the Einstein equations, and effective contributions from matter in vacuum of the form of Eq. (16). If we can compute the contributions of matter, then we can choose the value of the bare cosmological constant to match observations.

For a scalar field with action

$$S = -\int d^4x \sqrt{-g}\left(\frac{1}{2}g^{\mu\nu}\partial_\mu\phi\partial_\nu\phi + V(\phi)\right), \tag{17}$$

in vacuum we find

$$\langle T_{\mu\nu}\rangle = -V(\phi_{\text{min}})g_{\mu\nu}. \tag{18}$$

The potential energy stored in the field at the minimum of its potential behaves as an effective cosmological constant

$$\Lambda_{\text{eff}}^2 = -\frac{V(\phi_{\text{min}})}{M_P^2}. \tag{19}$$

If the value of $V(\phi_{\min})$ changes over time, for example if the scalar field undergoes a phase transition, this contribution to the cosmological 'constant' will also change. The universe has gone through at least two phase transitions, the electroweak, and the QCD, during its history. During the electroweak phase transition the Higgs field goes through a phase transition which leads to a change in the vacuum expectation value (vev) of the Higgs, and a change in the contribution to the vacuum energy density such that

$$|\Delta\rho_{\text{vac}}| \sim m_h^2 v^2, \tag{20}$$

where $v$ is the Higgs vev today and $m_h$ its mass. It is possible to choose the bare cosmological constant to ensure that, when summed with the contribution from the Higgs potential, the observed cosmological constant is small either before the phase transition, or after. But not both. How to decide when it is natural to perform this tuning, and how large a tunining of the bare cosmological constant can be considered natural are the classical parts of the cosmological constant problem.

## 4.1 Quantum Zero Point Energy

The cosmological constant problem gets worse when we consider that the universe is not only described by classical physics, but should also be described by quantum field theory. Still thinking about our scalar field, we choose its potential to be

$$V(\phi) = \frac{1}{2}m^2\phi^2. \tag{21}$$

As $\phi$ is a free field we can Fourier expand it

$$\phi(t,x) = \frac{1}{(2\pi)^{3/2}} \int \frac{d^3\vec{k}}{\sqrt{2\omega}} \left( c_k e^{-i\omega t + i\vec{k}\cdot\vec{x}} + c_k^\dagger e^{i\omega t - i\vec{k}\cdot\vec{x}} \right), \tag{22}$$

where $\omega^2 = k^2 + m^2$.

Substituting into the expression for the energy momentum tensor we find that

$$\langle\rho\rangle = \frac{1}{(2\pi)^3} \times \frac{1}{2} \times \int d^3k\,\omega(k), \tag{23}$$

$$\langle p\rangle = \frac{1}{(2\pi)^3} \times \frac{1}{6} \times \int d^3k\,\frac{k^2}{\omega(k)}, \tag{24}$$

but both of these integrals blow up!

We clearly need to regulate these divergences, but we need to be careful. If we just impose a hard cut off at the energy scale $M$ we find

$$\langle\rho\rangle = \frac{M^4}{16\pi^2}, \tag{25}$$

$$\langle p\rangle = \frac{1}{3} \times \frac{M^4}{16\pi^2}, \tag{26}$$

which gives an equation of state of $w = 1/3$. This makes the vacumm energy look like radiation, and not a cosmological constant.

In fact the problem here is that our regulation scheme breaks Lorentz invariance. To see what happens if we use a scheme which respects Lorentz invariance we instead try using dimensional regularization. This gives

$$\langle\rho\rangle = \frac{\mu^4}{2(4\pi)^{(d-1)/2}} \frac{\Gamma(-d/2)}{\Gamma(-1/2)} \left(\frac{m}{\mu}\right)^d, \tag{27}$$

$$\langle p\rangle = -\frac{\mu^4}{2(4\pi)^{(d-1)/2}} \frac{\Gamma(-d/2)}{\Gamma(-1/2)} \left(\frac{m}{\mu}\right)^d, \tag{28}$$

where $\mu$ is the regularization scale. So this time we find an equation of state with $w = -1$. If we subtract the pole in $\Gamma(-d/2)$ then we find

$$\langle \rho \rangle = -\langle p \rangle = \frac{m^4}{64\pi^2} \ln\left(\frac{m^2}{\mu^2}\right). \tag{29}$$

So the amount of vacuum energy scales with the mass of the heaviest particle in our theory. The heaviest observed particle to date is the top quark with a mass of 173 GeV. So our calculation predicts a contribution to the cosmological constant from the quantum zero point energy that is significantly larger than the observed energy density corresponding to the cosmological constant today which is $\sim (10^{-3} \text{ eV})^4$! It seems that the cosmological constant needs to be carefull fine-tuned to almost (but not quite) cancel the quantum zero-point energy.

## 5 Why Extend Gravity?

- **Why not?** We have argued in Section 2 that if the metric is the only field in the gravitational sector, then General Relativity is the unique theory of this field arising from an action principle in four space-time dimensions with second order equations of motion. However we could choose to break the assumptions that underlie Lovelock's theorem, and if we do there could be lots of interesting new phenomenology to study!

- **Dark energy and the cosmological constant problem.** As we have seen, the only option within standard physics to explain the observed acceleration of the expansion of the universe is a cosmological constant which needs significant fine tuning to match observations.

- **UV completion.** We can write a low energy effective field theory for gravity, however this theory is not UV complete. One way of addressing this could be through modifications of gravity.[5]

- **Parameterising deviations.** To fully test general relativity it is necessary to be able to predict which observables would differ in alternative theories, and to be able to parametrise the size of these deviations allowed by current observations.

## 6 How to Extend Gravity

There are many (many, many) ways to extend gravity (see Ref. [7] for a good review). What we will discuss here is retaining Lorentz invariance and universal coupling, but adding in additional fields, specifically an additional scalar field. We have already seen how to add a scalar field to the matter sector, what does it mean to add a scalar in the gravitational sector?

The way we will introduce our scalar modification here is to couple it non-minimally to gravity through a function $A(\phi)$, so that the gravitational action is

$$S = \int d^4x \sqrt{-g} \left( \frac{M_P^2}{2} A^2(\phi)R - \frac{1}{2} g^{\mu\nu}\nabla_\mu\phi\nabla_\nu\phi - V(\phi) + \mathcal{L}_m(g_{\mu\nu}, \psi_i) \right), \tag{30}$$

we can think of this as making Newton's constant (or equivalently the Planck mass) dependent on the scalar field. We remind the reader that $\mathcal{L}_m$ is the Lagrangian for the matter fields $\psi_i$.

---

[5]Although this is a reason to extend or modify gravity, it is unclear whether such modifications will also solve the cosmological constant problems or explain dark energy.

The theory expressed in the form of Eq. (30) is known as the Jordan frame description. We will generally assume that the scalar field $\phi$ is light in cosmological environments so that it is a relevant and dynamical degree of freedom on large scles in the late universe.

There is an equivalent description of this theory, known as the Einstein frame, which we find if we do the field redefinitions $\tilde{g}_{\mu\nu} = A^2(\phi)g_{\mu\nu}$ and

$$\left(\frac{d\tilde{\phi}}{d\phi}\right)^2 = \frac{1}{A^2}\left(1 + 6M_P^2\left(\frac{dA}{d\phi}\right)^2\right), \tag{31}$$

which results in the action

$$S = \int d^4x\sqrt{-\tilde{g}}\left(\frac{M_P^2}{2}\tilde{R} - \frac{1}{2}\tilde{g}^{\mu\nu}\tilde{\nabla}_\mu\tilde{\phi}\tilde{\nabla}_\nu\tilde{\phi} - \tilde{V}(\tilde{\phi}) + \mathcal{L}_m(A^{-2}(\phi(\tilde{\phi}))\tilde{g}_{\mu\nu}, \psi_i)\right), \tag{32}$$

where $\tilde{V} = \sqrt{\tilde{g}/g}V$, and $\tilde{\nabla}_\mu$ is the covariant derivative with respect to the metric $\tilde{g}_{\mu\nu}$.

For specific choices of $A(\phi)$ and $V(\phi)$ there is a third way of framing these theories as $f(R)$ theories of gravity. An $f(R)$ theory of gravity replaces the Ricci scalar in the Einstein-Hilbert action for General Relativity with a general function of the Ricci scalar. $f(R)$ theories of gravity are a rare example of a theory where higher derivative terms do not give rise to ghost instabilities, for a full discussion see Ref. [15]. The higher derivative terms do lead to a new degree of freedom, which can be recast as a scalar field [16], but this scalar mode is stable.

# 7 How to Drive Accelerated Expansion

## 7.1 Quintessence

As stated above, an unusual substance with an equation of state $w = -1$ can mimic a cosmological constant and drive an accelerated expansion. This can be achieved with a scalar field with the action

$$S = \int d^4x\sqrt{-g}\left(-\frac{1}{2}(\nabla\phi)^2 - V(\phi)\right). \tag{33}$$

By varying Eq. (33) with respect to $\phi$, and assuming that space-time is described by an FRW metric with scale factor $a$, we find that the background cosmological evolution is

$$\ddot{\phi} + 3H\dot{\phi} + \frac{dV}{d\phi} = 0, \tag{34}$$

and the components of the energy momentum tensor are

$$\rho = \frac{\dot{\phi}^2}{2} + V(\phi), \tag{35}$$

$$p = \frac{\dot{\phi}^2}{2} - V(\phi). \tag{36}$$

We see that the equation of state of this fluid can approach $w = -1$ if we are in a 'slow roll' regime where $\dot{\phi}^2 \ll V(\phi)$ (note we get accelerated expansion as long as $w < -1/3$).

We can get this slow-roll behaviour in a few different ways, particularly if the field is settling into the minimum of its potential, or if Hubble friction stops the field from rolling down its potential. A common choice of potential which allows Hubble friction (the $3H\dot{\phi}$ term in Eq. (34)) to stop the field at late times, known as 'freezing', is an inverse power law $V(\phi) = \Lambda^5/\phi$.

A quintessence field has the advantages that there can exist tracking solutions which can help solve the coincidence problem - why does the cosmological constant term come to dominate the evolution of the universe around the time of the formation of the solar system [6]? The existence of scaling solutions can also remove dependence on initial conditions [6]. However it doesn't help to answer the question of why the cosmological constant is not huge, and quintessence models contain an effective cosmological constant, of the form in Eq. (19), in the choice of value of the scalar potential $V(\phi)$ today, so the cosmological constant problems discussed in section 4 remain.

## 7.2 Self-Acceleration

For scalar field theories there is an alternative way to drive an accelerated expansion known as self-acceleration. Thinking about the scalar modification of gravity that we introduced earlier in Eq. (30), we had two descriptions, a 'Jordan frame' where the scalar couples explicitly to the metric, and an 'Einstein frame' where the scalar couples explicitly to matter, which are related by field redefinitions. Self-acceleration is the idea that the scale factor in the Jordan frame will accelerate but the expansion in the Einstein frame will not accelerate, even in the absence of a cosmological constant in either frame. If the calculations are done carefully, observables are the same whichever frame we calculate in (they should not be changed by field redefinitions!) however, implicitly, we normally do cosmological analysis in the Jordan frame (as we assume that particle masses are constants). The Jordan frame accelerated expansion comes entirely from the conformal transformation between the metrics and the dynamics of the scalar field.

The Jordan and Einstein scale factors are related by $a_J = Aa_E$. Comparing Friedmann equations (this discussion follows [17]) we can show that

$$a_J \ddot{a}_J - a_E \ddot{a}_E = \left( \frac{A'}{A} \right)', \tag{37}$$

where a dot is a derivative with respect to proper time in the frame of interest ($\dot{a}_E$ is the derivative of the Einstein frame scale factor with respect to the Einstein frame proper time), and a prime is a derivative with respect to conformal time (which is the same in both frames). If the Einstein frame scale factor is not accelerating then we must have

$$a_J \ddot{a}_J \le \left( \frac{A'}{A} \right)', \tag{38}$$

implying that $1 \lesssim \Delta A / A$ over a (Jordan frame) Hubble time. Therefore the scalar field has to evolve significantly to drive self-acceleration.

# 8 Scalar Forces and Screening Mechanisms

In these lecture notes we have previously considered scalar field extensions of gravity. These scalar fields interact with matter, and this can be seen directly in the Einstein frame in Eq. (30). If a scalar field couples to Standard Model matter it will mediate a new force. The force will be long range if the scalar is light. Experiments constrain long range forces to have couplings $\sim 10^5$ times weaker than gravity [12]. This means introducing an energy scale five orders of magnitude above the Planck scale. If we don't want to introduce another fine tuning, or contemplate physics above the Planck scale, one approach is to consider scalar fields which have screening mechanisms to suppress their fifth forces in the vicinity of large dense objects but still allow for significant deviations from general relativity on cosmological scales. In this section we will see how the fifth forces arise, and then how screening can dynamically suppress the fifth force in certain circumstances.

## 8.1 Scalar Forces

We will now compute the tree level 2-2 particle scattering interaction by exchange of a light scalar. This section follows the discussion in the textbook by Peskin and Schroeder [18], and as a result we change the sign convention of our metric in this section to $(+,-,-,-)$. We start from a Lagrangian

$$\mathcal{L} = \frac{1}{2}(\partial_\mu \phi)^2 - \frac{1}{2}m_\phi^2 \phi^2 + \bar{\psi}(i\gamma^\mu \partial_\mu - m)\psi + g\bar{\psi}\psi\phi. \tag{39}$$

The scalar propagator is

$$\frac{i}{q^2 - m_\phi^2 + i\epsilon}. \tag{40}$$

The fermion propagator is

$$\frac{i(\not{p} + m)}{p^2 - m^2 + i\epsilon}, \tag{41}$$

and the vertex contributes

$$-ig. \tag{42}$$

In order to compare this with local experiments testing gravity, we want to work out this interaction in the non-relativistic limit $p = (m, \vec{p})$ and $k = (m, \vec{k})$, where the three-momenta are small, and $(p' - p)^2 = -|\vec{p}' - \vec{p}|^2 + \mathcal{O}(\vec{p}^4)$. The external fermion is

$$u^s(p_0) = \sqrt{m}\left(\begin{array}{c} \xi^s \\ \xi^s \end{array}\right), \tag{43}$$

where $\xi$ is a 2 component spinor, and the factor of $\sqrt{m}$ is a convenient normalization such that $\bar{u}^r u^s = 2m\delta^{rs}$.

Now we can compute the scattering amplitude of our Feynman diagram

$$i\mathcal{M} = (-ig^2)\left(\bar{u}(p')u(p)\frac{i}{(p'-p)^2 - m_\phi^2}\bar{u}(k')u(k)\right) \tag{44}$$

$$\approx -g^2\left(2m\frac{i}{-|\vec{p}' - \vec{p}|^2 - m_\phi^2}2m\right), \tag{45}$$

$$i\mathcal{M} \approx \frac{4im^2 g^2}{|\vec{p}' - \vec{p}|^2 - m_\phi^2}, \tag{46}$$

where we have used the non-relativistic approximation in the second line.

We can compare this with non-relativistic quantum-mechanics governed by the Schrodinger equation

$$-\frac{\hbar^2}{2m}\nabla^2 \psi + V\psi = 0. \tag{47}$$

If a particle with average momentum $\hbar\vec{k}$ is incident on a potential $V$, the scattering amplitude is defined as the coefficient of the outgoing wave in the asymptotic solution.

If we assume that scattering is weak and the total wavefunction is approximately the incident wave function

$$\langle p'|iT|p\rangle = -i\tilde{V}(q)(2\pi)\delta(E_{\vec{p}'} - E_{\vec{p}}), \tag{48}$$

where $\vec{q} = \vec{p}' - \vec{p}$ and $\tilde{V}$ is the Fourier transformed potential.

In field theory

$$\langle in|iT|out\rangle = (2\pi)^4 \delta^{(4)}(k_{in} - k_{out})i\mathcal{M}, \tag{49}$$

so we identify

$$\tilde{V}(\vec{q}) = -\frac{g^2}{|q|^2 + m_\phi^2}\,, \tag{50}$$

where we have had to divide by $1/(2m)^2$ to convert from relativistic to non-relativistic normalizations.

Inverting this Fourier transform (close the integration contour with a semi-circle in the upper half of the complex plane) we find

$$V(r) = -\frac{g^2}{4\pi}\frac{1}{r}e^{-m_\phi r}\,. \tag{51}$$

## 8.2 Universally Coupled Scalars

We now return to the universally coupled scalar field introduced earlier. Matter fields move on geodesics of the Jordan frame metric

$$g_{\mu\nu} = A^{-2}(\phi)\tilde{g}_{\mu\nu}\,, \tag{52}$$

where $\tilde{g}_{\mu\nu}$ is the Einstein frame metric.

To understand how the scalar field affects matter, we work with a simplified situation, assuming that space-time is flat in the Einstein frame, $\tilde{g}_{\mu\nu} = \eta_{\mu\nu}$ and that the 'coupling function' is $A^2(\phi) \approx (1 - \phi/M)$. The motion of a matter particle, with position $X^\mu$, is governed by the geodesic equation

$$\frac{\partial^2 X^\nu}{\partial \lambda^2} + \Gamma^\nu_{\mu\rho}\frac{\partial X^\mu}{\partial \lambda}\frac{\partial X^\rho}{\partial \lambda} = 0\,. \tag{53}$$

The four velocity $u^\mu$ is normalised such that

$$\eta_{\mu\nu}u^\mu u^\nu = -1\,, \tag{54}$$

and the particle's acceleration is

$$a^\mu = u^\nu\nabla_\nu u^\mu\,. \tag{55}$$

By transforming the quantities in the geodesic equation into the Einstein frame, and defining a new four velocity $\tilde{u}^\mu$ (normalised such that $\tilde{g}_{\mu\nu}\tilde{u}^\mu\tilde{u}^\nu = -1$) and acceleration $\tilde{a}^\mu = \tilde{u}^\nu\tilde{\nabla}_\nu\tilde{u}^\mu$, we find

$$\tilde{a}^\nu = \tilde{u}^\mu\tilde{\partial}_\mu\tilde{u}^\nu = -\frac{\phi_{,\tilde{\nu}}/M}{1 + \phi/M}(3\tilde{u}^\mu\tilde{u}^\nu + \eta^{\mu\nu})\,, \tag{56}$$

where $\phi_{,\tilde{\nu}} = \partial\phi/\partial\tilde{x}^\nu$. If we consider a static, spherically symmetric situation such that $\tilde{u}^\mu = (1,\vec{0})$ then we find that

$$\tilde{a}_r = -\frac{\phi_{,r}}{M}\,, \tag{57}$$

to first order in $\phi/M$, where the derivative of $\phi$ is with respect to the Einstein frame radial coordinate.

## 8.3 Scalar Field Around a Source

We take the following Lagrangian

$$\mathcal{L}_\phi = -\frac{1}{2}(\partial\phi)^2 - \frac{1}{2}m^2\phi^2 + \mathcal{L}_m(\psi_i,(1 + \phi/M)g_{\mu\nu})\,, \tag{58}$$

and the overall energy-momentum tensor is

$$T_{\mu\nu} = -\frac{2}{\sqrt{-g}}\frac{\delta\mathcal{L}_m}{\delta g^{\mu\nu}} = -\frac{2}{\sqrt{-g}}\left(1-\frac{\phi}{M}\right)\frac{\delta\mathcal{L}_m}{\delta\tilde{g}^{\mu\nu}}. \tag{59}$$

The equation of motion for $\phi$ is then

$$\Box\phi - m^2\phi - \frac{1}{M\sqrt{-g}}\frac{\delta\mathcal{L}_m}{\delta\tilde{g}^{\mu\nu}}g^{\mu\nu} = 0, \tag{60}$$

$$\Box\phi - m^2\phi - \frac{1}{M}g^{\mu\nu}\left(-\frac{T_{\mu\nu}}{2(1-\phi/M)}\right) = 0, \tag{61}$$

$$\Box\phi - m^2\phi - \frac{1}{2M}T^{\mu}_{\mu} = 0. \tag{62}$$

If the source is static, non-relativistic and spherically symmetric we can write the energy momentum tensor as $T^{\mu}_{\mu} = \text{diag}(-\rho(r), \vec{0})$, so that the equation of motion is

$$\Box\phi = m^2\phi + \frac{1}{2M}\rho(r). \tag{63}$$

Now if we assume that the source has mass $M_s$, constant density $\rho$ and radius $R$ then

$$\phi'' + \frac{2\phi'}{r} - m^2\phi = \frac{1}{2M}\rho\Theta(R-r), \tag{64}$$

we can solve this by finding solutions for $r < R$ and $r > R$ and then imposing that $\phi$ and $\phi'$ are continuous at the surface of the source. We also impose that the field is regular at the origin and decays to zero at infinity. This becomes

$$\phi = \frac{\rho}{2Mm^3}\left(\frac{\sinh mr}{r} - m\right), \quad r < R, \tag{65}$$

$$\phi = \frac{1}{2M}\frac{M_s}{8\pi}\frac{e^{m(R-r)}}{r}, \quad r > R, \tag{66}$$

which has the form of the Yukawa potential.

The force experienced by a test particle is $F_\phi = \nabla\phi/M$ (see Eq. (57)), and so we can interpret the field value (divided by $M$) as the corresponding potential. The Yukawa force is therefore

$$F_{\text{Yuk}} = \frac{M_s(mr-1)}{16\pi M^2 r^2}e^{m(R-r)}. \tag{67}$$

Inside the Compton wavelength of the scalar field, where $e^{-mr} \approx 1$, we find

$$F_{\text{Yuk}} = -\frac{1}{2M^2}\frac{M_s}{8\pi r^2}. \tag{68}$$

## 8.4 Screening Around a Source

### 8.4.1 The Chameleon Model

We take the following Lagrangian, where we have chosen an inverse power law potential inspired by quintessence models

$$\mathcal{L}_\phi = -\frac{1}{2}(\partial\phi)^2 - \frac{\Lambda^5}{\phi} + \mathcal{L}_m(\psi_i, (1+\phi/M)g_{\mu\nu}). \tag{69}$$

The corresponding equation of motion for the chameleon field is

$$\Box\phi = V_{\text{eff}}(\phi), \tag{70}$$

where, in the presence of a non-relativistic background matter density $\rho$, the behaviour of the field is governed by an effective potential

$$V_{\text{eff}}(\phi) = \frac{\Lambda^5}{\phi} + \frac{\phi\rho}{M}. \tag{71}$$

For a given $\rho$ the minimum of the effective potential is

$$\phi_{\text{min}} = \left(\frac{\Lambda^5 M}{\rho}\right)^{1/2}, \tag{72}$$

and the mass of small fluctuations around this minimum is

$$m_{\text{min}}^2 = 2\Lambda^5 \left(\frac{\rho}{\Lambda^5 M}\right)^{3/2}. \tag{73}$$

It is therefore possible that the field behaves very differently inside and outside a compact source. Screening of the fifth force occurs if the field is so massive inside the source that there is a region inside the source where the field is essentially constant, and so no gradients of the field are built up. We call the radius of this region $R_*$. For $r < R_*$ we will assume that the field is constant and at the minimum of the effective potential. For $R_* < r < R$ we assume the potential is well approximated by $V_{\text{eff}} \approx \rho\phi/M$. Then outside the source we Taylor expand the potential around its minimum to approximate $V_{\text{eff}} \approx (1/2)m_\infty^2(\phi - \phi_\infty)^2$ where $\phi_\infty$ and $m_\infty$ are the minimum of the potential and the mass of small fluctuations in the background.

Constructing the field profile as before and imposing continuity of the field and its first derivative at $R_*$ and $R$ we find

$$\phi = \phi_{\text{in}}, \quad r < R_*, \tag{74}$$

$$\phi = \phi_{\text{in}} + \frac{\rho_{\text{in}}r^2}{6M}\left(1 - \frac{3R_*^2}{r^2} + \frac{2R_*^3}{r^3}\right), \quad R_* < r < R, \tag{75}$$

$$\phi = \phi_\infty - \frac{\rho_{\text{in}}R^3}{3M}\left(1 - \frac{R_*^3}{R^3}\right)\frac{e^{m_\infty(R-r)}}{r}, \quad r > R, \tag{76}$$

and the position of the surface $R_*$ is determined by

$$1 - \frac{R_*^2}{R^2} = \frac{2M}{\rho_{\text{in}}R^2}(\phi_\infty - \phi_{\text{in}}), \tag{77}$$

we see that if $R_*$ is close to $R$ the field in the exterior of the source is approximately constant.

We can take the ratio of the chameleon screened force to the unscreened Yukawa force with the same mass in the background to find

$$\frac{F_{\text{cham}}}{F_{\text{Yuk}}} = 2\left(1 - \frac{R_*^3}{R^3}\right) \approx \frac{3M}{\rho R^2}(\phi_\infty - \phi_{\text{in}}), \tag{78}$$

where $F_{\text{Yuk}}$ is as in Eq. (67), with $m = m_\infty$. If a source has a thin-shell, such that there is a non-zero $R_*$ which satisfies Eq. (77), then the corresponding scalar mediated fifth force will be suppressed.

### 8.4.2 Cubic Galileon

In this section we consider an example where screening of the fifth force arrises from a modifcation of the kinetic term for the scalar field

$$\mathcal{L} = \frac{1}{2}(\partial\phi)^2 + \frac{c_3}{\Lambda^3}\Box\phi(\partial\phi)^2, \tag{79}$$

again we couple to matter through a linear $\phi/M$ coupling. Despite higher order derivative terms in the Lagrangian, we find that the equations of motion are at most second order in derivatives

$$\Box\phi + \frac{c_3}{\Lambda^3}[(\Box\phi)^2 - \partial_\mu\partial_\nu\phi\partial^\mu\partial^\nu\phi] = \frac{\rho}{M}. \tag{80}$$

Taking the source to be spherically symmetric and of constant density as before we find

$$\frac{1}{r^2}\frac{\partial}{\partial r}\left(r^3\left[\left(\frac{\phi'}{r}\right) + \frac{c_3}{\Lambda^3}\left(\frac{\phi'}{r}\right)^2\right]\right) = \frac{\rho}{M}\Theta(R-r), \tag{81}$$

which has solutions

$$\phi' = \frac{\Lambda^3 r}{2c_3}\left(-1 + \sqrt{1 + \frac{4c_3\rho}{3M\Lambda^3}}\right), \quad r < R, \tag{82}$$

$$\phi' = \frac{\Lambda^3 r}{2c_3}\left(-1 + \sqrt{1 + \frac{4c_3\rho}{3M\Lambda^3}\frac{R^3}{r^3}}\right), \quad r > R. \tag{83}$$

When $R < r \ll R_V$ where $R_V^3 = c_3 M_s/\pi M\Lambda^3$ we find that the ratio of the screened to unscreened scalar forces is

$$\frac{F_{\text{gal}}}{F_{\text{unscreen}}} = 2\left(\frac{r}{R_V}\right)^{3/2}, \tag{84}$$

where $F_{\text{unscreen}}$ is the unscreened Yukawa force of Eq.(67) with $m = 0$. We see that inside the Vainstein radius the scalar mediated force is suppressed.

## 9 Summary

Solving the cosmological constant problem and explaining the accelerated expansion of the universe by other means is hard, and there are no widely accepted solutions. Even if we assume that some, as yet unknown, mechanism sets the observed cosmological constant to zero, it is still a challenge to explain the observed accelerated expansion without coming into conflict with other measurements. But there are many avenues still to explore, for example whether there exist viable theories which self-accelerate the universe that do not conflict with any other observational test. What makes this problem even more interesting is that the energy scale associated with it is a very accessible one, being roughly that of neutrino masses, and a distance scale of roughly 0.1 mm. This is a very well tested experimental regime.

There are many topics in this area that we have not touched on in these lectures. One significant example is the constraints that come from cosmological observations and also observations of gravitational waves. In the context of scalar tensor theories of gravity, a very nice review of these constraints can be found in this Reference by Johannes Noller [19].

## Acknowledgements

**Funding information** CB is supported by a Research Leadership Award from the Leverhulme Trust and a Royal Society University Research Fellowship.

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
