# Peer review of "A brief introduction to extended gravity and connections to dark energy: Illustrated with scalar field examples"

_SciPost Physics Lecture Notes, doi:SciPost Phys. Lect. Notes 41 (2022)_

## Round 1 · Referee Report · Anonymous (Referee 1) · 2021-10-1

Strengths

  1. very pedagogical introduction to the cosmological constant problem
  2. very pedagogical introduction to a non-minimal scalar coupling extension to GR
  3. a lot physically important concepts introduced in a clear way

Weaknesses

  1. not a real weakness but given the nature of the notes, being a summary of lectures, some elements are introduced in a rather fast and sometimes oversimplified fashion, but this is likely just related to the nature of the invited publication.

Report

Article meets journal requirements and is acceptable for publication

Requested changes

  1. the title and abstract of the paper suggest the lectures will be generically on "Extended Gravity", while the paper really focuses on a specific model. The introduction clearly motivates the reasons why this specific model is been studied instead of providing a comprehensive review, but it would useful to clarify the title and abstract accordingly.

  2. section 4 of the lectures correctly distinguishes between the classical and quantum aspects of the cosmological constant problems, but in listing various approaches in paragraph 3 of the introduction, it would be beneficial in clarifying which aspects of those cosmological constant problems the models intend to "solve"

  3. Related to the previous point, at the beginning of the 4th paragraph of the introduction, it states that "In these lectures I will focus in particular on the possibility that the explanation for how the universe is evolving is due to a modification of gravity." but it would be beneficial to clarify what aspects of the cosmological constant problems this approach is expected to tackle.

  4. Within string theory, solutions of the CC problem do not really rely on extra dimensions nor modifications of gravity but rather on anthropic principles

  5. It is unclear why it would be stated that giving the graviton a mass would be in tension with observations, this statement seems to be making some implicit assumptions.

  6. Just before eq. (1), there are quite a few other assumptions that go in deriving the uniqueness of GR+CC

  7. Since everything else has been introduced with some precisions, it would useful to indicate what T is eq.(5) (ie relate to eq.(13)) and what \rho and p stand for in eqns (7) (8) and (9)

  8. Einstein's equations have been derived with Mpl in section 2, but G is used in section 3, indicating the relation between those would be useful

  9. \sqrt{-g}L_m instead of S_m in eq. (13)

  10. Below eq.(14), the notion of Lorentz invariance is introduced on a generic (non-Minkowski) metric g_{\mu\nu}, what is meant by Lorentz invariance at level should be clarified.

  11. The 1st point of section 5, "Why Not?" seems to contradict the arguments given in section 2, where it was stated that (2) was the "only" thing we could have

  12. In going from eq.(26) to (28) it would be relevant to understand how L_m couples

  13. The sentence at the end of section 6 `` The gravitational instability of general relativity means that the new scalar mode does not introduce a ghost instability" may not be necessarily obvious to everyone.

  14. The discussions in sections 8.1 and 8.3, 8.4 are very interesting and useful but a better connection between both results would be useful. For instance it would be useful to indicate which F_{Yukawa} is used in eq.(72) and F_{unscreen} in eq.(78)

  15. It is unclear how (74) is connected to framework introduced in sec.6

---

## Round 1 · Referee Report · Anonymous (Referee 2) · 2021-11-2

Report

These are lecture notes on modified gravity in connection with dark energy. I find them very clearly written, in line with the current activity of the field and well focused. I only have a few minor comments before recommending them for publication.

Requested changes

1- In Sec. 5, among the motivations for extending gravity the author could add the need of parametrising deviations from standard general relativity with the goal of constraining them with future observations, analogously to what one does with precision tests at LHC.

2- In Sec. 6, I do not understand the last sentence ("The gravitational instability...").

3- In Sec. 8.1 the signature of the metric changes, which is understandable because the calculation is taken from a particle physics book. The author could point this out with a comment.

4- I agree with the calculation of Sec. 8.2 but a source of confusion comes from the fact that previously (see Sec. 6) the tilde was used to denote the metric in the Einstein frame. Matter particles move on geodesics in the Jordan frame so I would have expected not to see tildas in eq. (49) and to see them in eq. (52) and (53). It is probably just a question of notation but the author could clarify this point.

5- The end of Secs. 8.4.1 and 8.4.2 is a bit dry. The author could add a line to comment the final equations of these subsections, which are relevant results of these notes.

6- Finally, for the first sentence of the conclusion I would find more appropriate something like "...solving the cosmological constant and explaining the accelerated expansion of the universe by other means...". My impression is that these notes focus more on phenomenology of modified gravity than on the cosmological constant problems.

7- There are a few minor typos, such as - space time -> spacetime - which ever -> whichever

---

## Round 1 · Referee Report · Anonymous (Referee 3) · 2021-11-5

Report

Overall, I believe that this article mentions some important topics in gravity and cosmology, but I find the discussion insufficient, confusing, and sometimes misleading or incorrect. I think that the best way for me to explain my reasoning is to start by looking at claims made in the Introduction and Summary.

Introduction: - "In these lectures I aim to demonstrate the problems with the cosmological constant and its solutions." - I don't believe that the cosmological constant (CC) problem is explained in this article. On page 5 there is a discussion of zero-point energies, but there is no explanation for why eq. (25) is a problem. I would expect the classic estimate of mismatched scales, but this is not shown. Fine tuning is mentioned two times, but the concept and the problems with it are never explained. It is said on page 7, "all of the cosmological constant problems remain," but it is not clear from this article what all of those problems are. Even more, it is not clear from this article what a proposed solution to the CC problem is, or how any of the discussions in the article relate to the CC problem.

  • "Gravity has been extremely well tested in the laboratory, and in the solar system." - This is true, but there is no connection drawn between this and the contents of the article. For example, the author describes screening and forces, but there is no discussion of how these are related to laboratory or solar system tests. The typical comment would be that a screening mechanism is needed so that gravity is modified on cosmological scales, but not on smaller scales where we have tests, but such a discussion is not present.

  • "We will see that we may want to modify gravity on long distance scales." - I do not believe that this article explains, or attempts to explain, why this is the case. There is mentioned in the Introduction and conclusion that GR is tested in the lab and solar system, but the main text does not seem to address this issue. See also point above.

Summary: - "The main message of these lectures has been that solving the cosmological constant problem is hard." - As explained above, this article does not mention why solving the CC problem is difficult. In fact, Sec. 7.2 is about self-acceleration which seems to hint at a possible solution (although this is not actually the case since the bare CC would still have to be zero). So, why doesn't self-acceleration work? How is this connected to the CC problem? How is self-acceleration different from the CC?

  • "... it is so difficult to construct new theories which pass all existing tests." - What tests? What is the connection between the scalar-field models presented and these tests? Do the models presented fail these tests, or do they succeed?

  • "Perhaps this is another indication that the solution must be non-linear, and that what we observe is a reprocessing of other more fundamental scales." I'm not sure about the meaning of this sentence. First, I assume the author is referencing the fact that non-linearities in the scalar-field action can induce Vainshtein screening, which is one way to attempt to evade local experiments [1304.7240, for example]. However, this is not discussed. The "reprocessing of other more fundamental scales" is also not explained.

General remarks: - page 4, "We can see now that it's possible to do this before or after a phase transition, but not both." I don't see where the role of phase transitions are explained, or any evidence to back up this claim.

  • On the top of page 6 it is mentioned that modifications of gravity could UV complete GR. This seems like a very strong statement, and I don't think that most people working on DE are trying to UV complete GR. The famous UV completion of GR is string theory, which I do not think is what the author means by modification of gravity in this context.

  • Around eqs. (26) and (28), the author describes a field redefinition to go from Jordan to Einstein frames. However, without including matter, these two frames are equivalent, as we know from the EFT of inflation [0709.0293] which does not include the $A^2 ( \phi )$ coupling in front of $R$ . Furthermore, the author says, "The gravitational instability of general relativity means that the new scalar mode does not introduce a ghost instability." I don't know what is meant by "gravitational instability" in this context, but generally the absence of ghosts depends on the details of the DE interactions, as happens for example in Horndeski and DHOST theories [1510.06930].

  • page 7 mentions a 'hidden' cosmological constant in quintessence models, but this is never explained.

  • In Sec. 7.2 it is claimed that "observables are the same which ever frame we calculate in (they should not be changed by field redefinitions!)." While this is true for the S-matrix, for example, this is not true of correlation functions, which are the normal observables computed in cosmology. The difference between the Jordan and Einstein frames is that in the Jordan frame, all matter is minimally coupled to gravity, and thus the equivalence principle is satisfied. The choice of the Jordan frame is a practical one based on the observation that the equivalence principle seems true. [1210.0201]

  • The Yukawa potential is derived using the Born approximation in the scattering of fermions exchanging a massive scalar, but the importance of the expression eq. (27) with respect to this article is never discussed. The same is true about other results (eqs. (53), (63), (72), and (78)).

---

## Round 2 · Author Response

I would like to thank the referees for their helpful and constructive comments. I have endeavoured to take all of their comments and suggestions into account in the revised manuscript.

---

## Round 2 · List of Changes

Changes have been made throughout the article to clarify points, and add references and additional discussion.

You are currently on this page

Resubmission scipost_202109_00007v2 on 13 December 2021

---

## Editorial Decision

published